# Diet Quality at 3 Years of Age Relates to Lower Body Mass Index but Not Lower Blood Pressure at 10 Years of Age

**DOI:** 10.3390/nu16162634

**Published:** 2024-08-09

**Authors:** Qihua Wang, Tian Xie, Xia Huo, Harold Snieder, Eva Corpeleijn

**Affiliations:** 1Department of Epidemiology, University Medical Center Groningen, University of Groningen, 9713 GZ Groningen, The Netherlands; q.wang@umcg.nl (Q.W.); h.snieder@umcg.nl (H.S.); 2Laboratory of Environmental Medicine and Developmental Toxicology, Guangdong Key Laboratory of Environmental Pollution and Health, College of Environment and Climate, Jinan University, Guangzhou 511443, China; xhuo@jnu.edu.cn

**Keywords:** DASH, MDS, LLDS, childhood BP, childhood BMI

## Abstract

A healthy diet prevents overweight problems and hypertension. We investigated the associations of a healthy diet with the body mass index (BMI) and blood pressure (BP) in early childhood. In the GECKO birth cohort, height, weight, and BP were measured at 5 and 10 years of age. Diet was evaluated at 3 years using three diet scores: the Dietary Approaches to Stop Hypertension (DASH), the Mediterranean Diet Score (MDS), and the Lifelines Diet Score (LLDS). Linear and logistic regression models assessed the associations of diet scores with the BMI and BP. Of the 1077 children included, 10.8% were overweight or obese at 5 years. That number was 16.5% at 10 years. In addition, 34.5% had elevated BP at 5 years. That number was 23.9% at 10 years. Higher DASH, MDS, and LLDS, which indicate healthier diets, were all associated with lower BMI z-scores at 10 years of age. Higher DASH is related to lower overweight risk at 10 years. None of the diet scores were associated with BP or elevated BP at either 5 or 10 years. Also, in an overweight subset, diet was not related to BP. A healthy diet in early childhood is related to children being less overweight but not having lower BP at 10 years of age.

## 1. Introduction

Childhood hypertension and overweight problems are global public health issues leading to long-term cardiovascular risk [1]. The global prevalence in children and adolescents was 4.0% for hypertension [2], 9.7% for prehypertension [2], and 14.8% for overweight problems [3]. Although many cohort studies show that hypertension, being overweight, or an increasing BMI trajectory during childhood predicts adulthood hypertension and cardiovascular diseases (CVD) [4,5]; intervention studies from early childhood are limited. Lifestyle and dietary consultations were reported by several studies to slightly contribute to improved cardiovascular health beginning from childhood, but few studies directly proved that long-term intervention or treatment of cardiovascular risk factors in childhood reduces the future risk of cardiovascular events [5]. It suggests the importance of early childhood in view of primordial prevention of CVD, which warrants additional focus.

A healthy diet is recommended by guidelines to prevent or reverse hypertension in adults and children, but the impact of different dietary patterns in children needs more research [6,7]. The Dietary Approaches to Stop Hypertension (DASH) diet was designed to decrease blood pressure (BP) and was shown to be effective in adults [7,8,9,10,11]. The Mediterranean diet, first studied in Greece and Southern Italy during the 1960s, improves the overall cardiometabolic risk profile and is associated with lower BP in adults [9,10,12]. The Lifelines Diet Score (LLDS) is a recently established diet score representing a dietary pattern adhering to the 2015 Dutch Dietary Guidelines [13]. It discriminated well between people with widely different intakes [13]. Higher LLDS scores, reflecting a healthier diet, were associated with lower BP, lower metabolic risk, and lower all-cause mortality in adults [14,15].

The associations of DASH and the Mediterranean Diet Score (MDS) with blood pressure or hypertension in childhood are inconsistent across studies, while the association between LLDS and BP in childhood has not yet been investigated. DASH was associated with lower BP and a lower prevalence of hypertension by cross-sectional and longitudinal studies in children and adolescents—healthy or ill [16,17,18,19,20,21,22]. DASH is effective in short-term and long-term clinical trials for decreasing BP in adolescents 10 to 18 years of age [23,24,25]. However, absent or opposite associations between DASH and childhood BP are also reported by cross-sectional and longitudinal studies [26,27,28,29]. Similarly, associations of the MDS with BP and hypertension were inconsistent in children and adolescents, indicating beneficial (2 cross-sectional studies), non-beneficial (1 cross-sectional study), and non-significant (3 cross-sectional studies and 1 longitudinal study) associations [30,31,32,33,34,35]. In contrast, a more consistent benefit of the MDS on BP in childhood was found in short-term and long-term clinical trials [36,37]. So far, there is limited research on LLDS and BP. Although higher LLDS values were associated with lower BP in adults, the relation with BP in children and adolescents has not yet been studied.

Similarly, well-known risk factors for hypertension—the BMI, being overweight, and obesity—and their association with diets have been investigated in many studies [38]. DASH has been associated with lower levels of the BMI [22] and weight gain [39], as well as people being less overweight or obese [16,40] and metabolically unhealthy obesity [29] in children older than 5 years in cross-sectional and longitudinal studies. In contrast, a few studies reported no association between DASH and the BMI or weight gain [20,25,41,42]. Similarly, the MDS was associated with lower levels of the BMI [34,36] and BMI gain [43], as well as people being less overweight or obese [34,37,44] in cross-sectional, longitudinal, and intervention studies, with a few studies showing inconsistent results [35,41,42,45]. So far, the only study on the LLDS in children showed that a higher LLDS is associated with a lower 7-year increase (from 3 to 10 years) in weight gain [46]. Further associations between the LLDS and BMI/obesity at other ages in childhood have not been reported.

Despite the existence of a large volume of studies that have assessed the influence of dietary patterns on BP and the BMI ranging from middle childhood to puberty, studies investigating this question in preschoolers younger than 5 years are limited. To fill this knowledge gap, we conducted a study from early to middle childhood to investigate the following three questions: (1) Is poor diet quality at 3 years related to a higher BMI and being overweight or obese at 5 and 10 years of age? (2) Is a poor diet quality at 3 years related to higher blood pressure and hypertension at 5 and 10 years of age? (3) Does the BMI influence the association between diet and blood pressure, and if so, how? We hypothesized that poor diet quality at 3 years of age may increase BP level and hypertension risk at 5 and 10 years of age and that the association depends on the BMI.

## 2. Materials and Methods

### 2.1. Study Design and Population

The GECKO Drenthe cohort study is a birth cohort focusing on early risk factors for the development of obesity in children. This cohort follows two-thirds of the children born in 1 year during 2006–2007 in Drenthe, a northern province in the Netherlands [47]. The children and their parents have been followed up during regular visits to the Well Baby Clinics during the first 4 years of life and thereafter via the school health services at 5/6 and 10/11 years [47]. The inclusion and exclusion criteria in the present study are shown in the flowchart (Appendix A). Among the 2842 active participants with informed consent in the GECKO cohort, 1077 individuals were included in the final analyses. The results of the attrition analysis are in the footnote of Appendix A. The study was conducted according to the guidelines of the Declaration of Helsinki and was approved by the Medical Ethics Committee of the University Medical Center Groningen. Written informed consent was obtained from parents. The cohort is registered at www.birthcohorts.net.

### 2.2. Outcomes

The main outcomes of interest were systolic blood pressure (SBP), diastolic blood pressure (DBP), elevated BP, BMI, and overweight issues.

BP at the age of 5–6 and 10–11 years was measured by trained preventive child healthcare nurses in 3 consecutive measurements with a digital automatic BP monitor (Omron M3 intellisense, OMRON Healthcare Co., Minato City, Japan). The child was sitting for 5 min before the BP was measured. The means of SBP and DBP were calculated from the 3 measurements and adjusted for cuff size where necessary. According to the 2017 Clinical Practice Guideline for Screening and Management of High Blood Pressure in Children and Adolescents, elevated BP was defined as ≥the age-, height- and sex-specific 90th percentile (or 120/80, whichever was lower), while hypertension was defined as ≥the age-, height- and sex-specific 95th percentile (or 130/80 whichever was lower) [48].

Height and weight at the ages of 5–6 and 10–11 years were measured by trained preventive child healthcare nurses according to standardized protocols, as described previously [49], using electronic scales without shoes or heavy clothing. BMI (kg/m^2^) was calculated at 5 and 10 years of age. The age- and sex- standardized BMI z-score was calculated using Growth Analyser software 3.5 (Dutch Growth Research Foundation, Rotterdam, The Netherlands), with the 1997 Dutch Growth study as the reference population [50]. According to the Extended International Obesity Task Force (IOTF) Body Mass Index Cutoffs from 2012, overweight, including obesity, was defined as BMI z-scores above 1.310 SD for boys and 1.244 SD for girls, while obesity was defined as BMI z-scores above 2.288 SD for boys and above 2.192 SD for girls [51].

### 2.3. Determinants

Dietary data over the past 4 weeks were collected using a food frequency questionnaire (FFQ) specific for children at 3 years of age. The FFQ was developed according to the Dutch National Food Consumption Survey 1997–1998 [52]. The FFQ contained 71 food products, with frequency categories from “never” to “6–7 times a week”. The units of the amounts were portions or common household measures. Parents were asked to measure the volume of glasses and cups used for different beverages. Verified and validated FFQ data were processed to calculate the daily intake of each food product (amount in portions ×portion size × weighting value/sum of weighting values × correction factor ×frequency/7 = daily intake in grams per day), after which daily energy and nutrient intake were calculated based on the Dutch food composition database (NEVO) of 2011 [53] (daily intake in gram per day × the amounts of energy and nutrients for per unit of each food product). The reliability of reported dietary intake was assessed using the Goldberg cutoff method and relied on the ratio of reported energy intake and basal metabolic rate [54,55,56]. Children with an energy intake/basal metabolic rate ratio below 0.87 or above 2.75 (N = 91) were excluded to limit bias through under- or over-reporting [54]. For more details, refer to TNO report V8643: Food frequency questionnaire on energy intake of children 2–12 years old: Verification and calculation procedures.

The DASH score was calculated according to the European ALPHABET Consortium (for pregnant women) [57] and LifeCycle Data Harmonization Protocol 2018 (for pregnant women and children older than 5 years of age) [58], containing 8 food groups (Appendix A). Daily intake of each food group and sodium was ranked into quintiles and given positive or reverse scores (from 0 to 4), summing up to a total score that ranged from 0 to 32.

The MDS was calculated according to the Identification and Prevention of Dietary- and Lifestyle-Induced Health Effects in Children and Infants Study (IDEFICS) (for children) [59], containing 7 food groups (Appendix A). The daily intakes in grams/1000 kcal of each food group and the ratio of saturated to unsaturated fats were expressed as a dichotomous ranking including high adherence (1) and low adherence (0), summing up to a total score that ranged from 0 to 7. High adherence to MDS was defined as an MDS higher than 3.

The LLDS was calculated based on 29 systematic reviews regarding associations of diet with chronic diseases, which the Dutch Health Council evaluated in the process of the development of the 2015 Dutch dietary guidelines (for adults) [13]. The daily intake in grams/1000 kcal of 11 instead of 12 food groups, excluding coffee for children, were calculated from FFQ data and expressed in a quintile ranking of the sum of daily intakes of each product (valued as 0 to 4), summing up to a total score ranging from 0 to 44 (Appendix A). A higher DASH score, MDS, or LLDS represents better diet quality.

### 2.4. Statistical Analyses

The associations of diet scores with continuous outcomes (SBP, BDP, BMI z-score) at 5 and 10 years were assessed using linear regressions and with dichotomous outcomes (elevated BP, overweight including obesity) at 5 and 10 years using logistic regressions. Diet scores were included in each model separately, both as continuous variables and categorical variables. As categorical determinants, DASH and LLDS were expressed in quintiles, with the highest quintile (best diet quality) as the reference level, while MDS was expressed as a dichotomy of high and low adherence. In the analyses, potential confounders were added to the crude model (model 1) in 3 steps (model 2, model 3, and model 4). In model 2, we adjusted for important basic covariates that affect outcome measurement, including age, sex, and height in the BP/elevated BP model and age and sex in the BMI/overweight model. In model 3, we additionally included socio-economic factors. To avoid overfitting, we only included one socio-economic factor with the largest coefficient based on the result of univariate models; these were maternal education level in the BP/elevated BP model and paternal education level in the BMI/overweight model. In model 4, we included any smoking during pregnancy, a common risk factor that may affect our outcomes based on existing evidence [60,61]. All tests were 2-sided, and *p* < 0.05 was used as the cutoff for statistical significance. Analyses were conducted using R version 4.0.5 and IBM SPSS Statistics 28 (SPSS, Chicago, IL, USA).

For the first research question, we investigate the associations between diet scores at 3 years of age and the BMI/overweight problems, including obesity at 5 and 10 years of age. For the second research question, we investigate the associations between diet scores at 3 years of age and BP/elevated BP at 5 and 10 years of age. For the third research question, we investigated the association of BMI z-score with BP and the associations of diet scores at 3 years of age with BP/elevated BP at 5 and 10 years of age only in children overweight including obesity at baseline (3 years of age). Considering the primary role of sodium in regulating BP and to avoid overestimated weight of sodium in the DASH pattern, we also investigated DASH without sodium instead of DASH as the determinant in the associations with BP/hypertension in sensitivity analyses.

## 3. Results

### 3.1. Demographic Characteristics

In total, a maximum of 1077 participants were included in the analysis (see flowchart Appendix A). Table 1 shows the characteristics of the study participants. Ages at outcome measurements were 5.83 (5.58, 6.00) and 10.58 (10.25, 10.83) years, and 48.7% of the participants were female.

### 3.2. Diet Scores and BMI/Overweight Issues

Overall, a better diet quality was related to a lower BMI and being less overweight or obese, mostly at 10 years of age (Figure 1B,D). An increase in any of the 3 diet scores on a continuous scale (DASH, MDS, and LLDS) at 3 years of age of 1 point was associated with a lower BMI z-score at 10 years of age (Figure 1B). In addition, an increase in DASH or MDS at 3 years of age was associated with lower overweight risk at 10 years of age (Figure 1D). However, when diet scores were assessed as categorical variables, the strongest associations were found for the DASH score (Figure 2).

Sensitivity analyses showed that DASH without sodium was not associated with BMI z-score at 5 years of age [B and 95%CI, model 4: −0.002 (−0.014, 0.010)], whereas the sodium score [median and range were 2 (0, 4), higher score represent lower sodium intake in mg per day] associated with lower BMI z-score at 5 years of age [B and 95%CI, model 4: −0.049 (−0.086, −0.012)]. Both DASH without sodium and isolated sodium score were associated with lower BMI z-score at 10 years of age [model 4 of DASH without sodium: B and 95%CI were −0.018 (−0.033, −0.002), model 4 of sodium score −0.056 (−0.104, −0.008)]. These results are consistent with our main analyses, which found that DASH was associated with a lower BMI z-score at 10 years but not at 5 years. It seems that sodium in DASH played a more important role than other food groups in the association of DASH on BMI z-score.

### 3.3. Diet Scores and BP/Elevated BP

As shown in Figure 3, none of the 3 diet scores were associated with SBP, DBP, or elevated BP at 5 or 10 years of age. Similarly, no significant associations were observed in quintiles of DASH or LLDS with BP or elevated BP at 5 or 10 years of age compared with the reference quintile, indicating the best diet quality, except the 2nd DASH quintile, indicating a relatively poor diet quality, which had a borderline significant association with lower SBP at 10 years of age (Figure 4). In the sensitivity analysis, neither DASH without sodium nor the isolated sodium score was associated with SBP or DBP at 5 or 10 years of age.

### 3.4. Role of BMI in the Association between Diet Scores and BP

Since blood pressure was related to a higher BMI (in the model adjusted for age, sex, height, maternal education level, any smoking during pregnancy, B_SBP~BMI_ = 2.092, B_DBP~BMI_ = 1.513 at 5 years, and 1.479, 1.044 at 10 years, all *p* < 0.001), we hypothesized that diet may have more impact on blood pressure in children with a compromised metabolic profile, e.g., in children with overweight including obesity. Therefore, we related diet to BP outcomes in an overweight (including obesity) subset of all children. Figure 5 shows that the associations between diet scores and BP in the overweight subset were similar as in all children: in children who were overweight at baseline (3 years of age), DASH and MDS were associated with lower SBP at 5 years only in the crude model, whereas MDS associated with lower SBP at 10 years only in the crude model.

## 4. Discussion

In the present study, we found that a healthy diet quality at 3 years of age relates to a lower BMI z-score and a lower risk of being overweight or obese but not to BP outcomes at 10 years of age. What is more, no association between diet quality and BP outcomes was found in a subset of children who are overweight or obese.

### 4.1. Diet Scores and BMI/Overweight Issues

We found associations of DASH and MDS with lower BMI z-score (and DASH also with fewer overweight issues) at 10 years of age. Previous longitudinal studies at similar ages showed both consistent [39] and inconsistent [41,42] results compared with the present study and reported no associations [35,41,42] more often than significant associations [43].

The association we found in the present study between a higher LLDS and a lower BMI z-score at 10 years of age is consistent with our previous study linking LLDS with a lower gain of weight and overweight incidence over a 7-year follow-up in childhood [46]. Our results indicated that a good diet quality at 3 years of age could benefit the BMI as shown over a 7-year follow-up in childhood. It could be explained by a sustained adherence to a healthy dietary pattern during childhood, based on the evidence that family diet and diet in early life were associated with diet at a later stage [62]. Alternatively, the dietary pattern at 3 years of age itself may have a long-term effect on the BMI z-score and on being overweight or obese at 10 years of age. The underlying mechanisms of dietary patterns as a whole decreasing BMI have not been well investigated, but the mechanisms may include reducing hunger, caloric restriction, and stimulation of dietary adherence [63]. Another thing that should be taken into consideration is that the prevalence of overweight or obese children in the present study was lower than that in European children in other studies, for example, 19.0% at preschool age in the ToyBox study [64]. One possible explanation is that the participants in the present study were children with normal birth outcomes, who were supposed to have a lower BMI z-score and overweight risk than children with poor birth outcomes. Sampling bias could explain the lower overweight prevalence at 10 years but not at 5 years in the present study because, according to our test, there was no sample bias at 5 years. So, we also suggested other explanations, including lower estimates due to the IOTF cutoffs.

In the present study, the associations of diet scores with the BMI z-score and with being overweight or obese were more significant at 10 years of age than at 5 years of age. One explanation might be that, for some individuals, the condition of being overweight was not yet developed at 5 years of age but was developed at 10 years of age. In addition, the childhood BMI has a curvilinear relation with age instead of a linear relation [51], leading to a relatively complex trajectory of associations.

Although the associations of diet scores, BMI z-scores, and being overweight or obese did not always show consistent trends from the 1st quintile to the 4th quintile, the 1st quintile indicating the poorest diet quality of DASH and LLDS was always the category with the strongest association with a higher BMI z-score and overweight incidence at 10 years of age. Overall, there is a convincing association between poor diet quality and being overweight.

### 4.2. Diet Scores and BP/Elevated BP

The lack of association between DASH and BP in the present study is consistent with several cross-sectional studies and longitudinal studies in children. Previous cross-sectional and longitudinal studies with a 4-year follow-up in children older than 6 years showed no benefit of DASH on BP, even though an association between DASH and cardiometabolic biomarkers had been observed [26,27]. However, observational studies with longer follow-ups, for example, 6-year [22] and > 10-year [19] follow-ups, found associations between DASH and lower BP. In line with this, associations between DASH and lower BP were found in intervention studies using either short-term (for example, 6-week [23] and 10-week [24]) or long-term follow-up (6 month [25]) in children with underlying diseases including metabolic syndrome [23], hemophilia [24], and elevated BP [25]. As for the association between MDS and BP, both significant [30,31] and absent [31,32,34] associations of MDS with lower BP or lower odds ratio of elevated BP in children were found in cross-sectional studies, with absences of associations being more predominant. A previous longitudinal study in adolescents of multiple ethnicities showed no association of MDS with BP [35], which is consistent with the present study. However, consistent associations were found in MDS intervention studies of 4 weeks [37] and 12 months [36]. To our knowledge, the present study is the first one to investigate the association of LLDS with BP in children. Although it is inconsistent with the study in adults showing that adherence to the highest tertile of LLDS is associated with lower SBP, the present study provides a novel perspective on this association in very early childhood.

One reason why diet quality was not related to BP outcomes in young children may be related to the mechanisms by which diet can affect blood pressure. Short-term effects may relate to a high intake of sodium, which increases BP, while a high intake of potassium [65], magnesium [66], and calcium [67] decreases BP both in the short- and long-term [68,69,70,71,72]. Short-term interventions of diet patterns decrease BP in adults and overweight children [73,74], but some of them are only effective in the short term but not in the long term [75,76]. It may be that we have not been able to detect a short-term effect of high salt intake because the day-to-day variation is high, and the FFQ captures dietary habits over the longer term. Long-term mechanisms include vascular remodeling and arterial stiffness, which take time to develop and may not occur yet in children, as opposed to elderly adults [77]. In that case, the 7-year follow-up in childhood in the present study is not long enough to observe the effects on the development of hypertension. To summarize, short-term effects of diet on decreasing BP are more often reported and more consistent than long-term effects [76,77]. Although we have not observed a long-term association (2-year or 7-year follow-up), there might be an unmeasured short-term association. Alternatively, changes in blood pressure may take a long time to manifest in response to dietary changes. On the other hand, the sampling methodology could partly account for variability in results and possible associations. To exclude prenatal factors as probable confounders relating to childhood BP in previous studies, we excluded children with premature birth or low birth weight. It made the present study focus on the major research questions but, at the same time, excluded possible associations of diet quality with BP through, or when interacting with, prenatal factors.

### 4.3. Role of BMI in Relationship between Diet and BP/Elevated BP

The association of the BMI with BP in the present study is in line with most previous studies, which found obese children were at higher risk of developing hypertension with multiple mechanisms potentially contributing, including activation of the renin–angiotensin–aldosterone system and the sympathetic nervous system, vascular dysfunction, other mineralocorticoid activity, and reduced kidney function [78,79]. However, we found no evidence of a stronger relationship between diet quality and BP in overweight children.

### 4.4. Strengths and Limitations

The present study has several strengths. First, our study has quite a long follow-up (from 3 years old to 10 years old), which is rare in previous studies in children. We investigated the associations of diet scores at 3 years with outcomes at both 5 years (preschoolers) and 10 years of age (middle childhood). Second, we highlight the significance of dietary analysis at the pattern level. Investigating diet quality as a whole instead of isolated nutrients is more important because the former is based on dietary recommendations and considers the imbalances among diet components. Third, 3 diet scores are investigated in the present study, supplying 3 different evaluation criteria of the diet quality of the same diet profile at the same time. In addition, the association of LLDS (a relatively newly established diet score) with BP in childhood has not been studied previously.

The limitations of the present study are as follows. First, the conclusion is not causal due to the observational design. Second, although many covariates have been included in our adjustments, residual confounding effects from unmeasured or unknown covariates cannot be completely ruled out. We have not adjusted physical activity in our models, but in a previous study in the same population, no association of early physical activity with BMI outcomes or BP outcomes was found [80]. We exclude individuals with adverse pregnancy outcomes at the start, so we have not included pregnancy outcomes as covariates. Third, the BP is the mean of three measurements in one visit instead of in separate visits; the same is true for the BMI measurements, so the definition of elevated BP/hypertension and overweight problems/obesity should be considered with caution as they are no clinical definitions. Fourth, only 1077 individuals who have normal birth outcomes among the whole cohort (N = 2842, including individuals with incomplete data, premature birth, or low birth weight) were included in the present study, which might have led to some bias.: The relationships between diet quality and BP or the BMI might be different for the participants with normal birth outcomes and the excluded participants who have poor birth outcomes. As a result, care should be taken when trying to extrapolate our results to other populations including participants with poor birth outcomes. Fifth, the MDS and LLDS calculations adapted to our population at a very young age, but the adaption to children of DASH should be taken into consideration. Although all three diet scores were based on an FFQ and a data process manual specific to children, the DASH calculation used food group criteria for pregnant women and school-aged children, with a daily intake in servings per day instead of adjusting for total energy intake. Differently, although the LLDS calculation was based on criteria for adults, we have excluded the coffee group in order to incorporate children’s diets, and the daily intake, which was adjusted for total energy intake, made up for the intake amount difference with adults. We suggested a further study assessing the accurate adaption of DASH to children younger than 5 years of age. Sixth, the FFQ data could have recall bias even though the data are validated. Under- or over-reporting has been excluded by the Goldberg cutoff method and relied on the ratio of reported energy intake and basal metabolic rate.

## 5. Conclusions

In conclusion, a better diet quality at 3 years of age relates to a lower BMI but not a lower BP at 10 years of age. These results were confirmed in overweight children and for elevated BP. Our findings suggest that a healthy diet in early childhood may prevent overweight problems in middle childhood, but the effect on BP needs further study with a longer follow-up.

## Figures and Tables

**Figure 1 nutrients-16-02634-f001:**
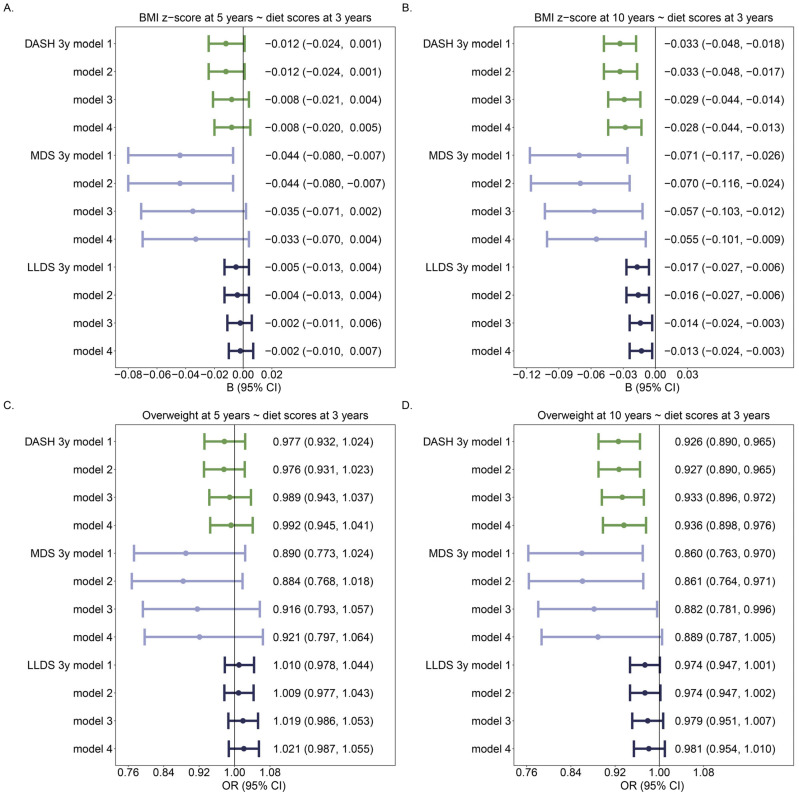
Associations of diet scores (per unit) are continuous with BMI z-score and overweight including obesity: (**A**) Association between diet scores at 3 years of age and BMI z-score at 5 years of age; (**B**) Association between diet scores at 3 years of age and BMI z-score at 10 years of age; (**C**) Association between diet scores at 3 years of age and overweight at 5 years of age; (**D**) Association between diet scores at 3 years of age and being overweight at 10 years of age. For each diet score, there were 4 models: Model 1: Crude model; Model 2: model 1 + age + sex; Model 3: model 2 + paternal education level; Model 4: model 3 + any smoking during pregnancy. The number of individuals was 971 in 5y models, and 959 in 10y models. Different colors identify different diet scores.

**Figure 2 nutrients-16-02634-f002:**
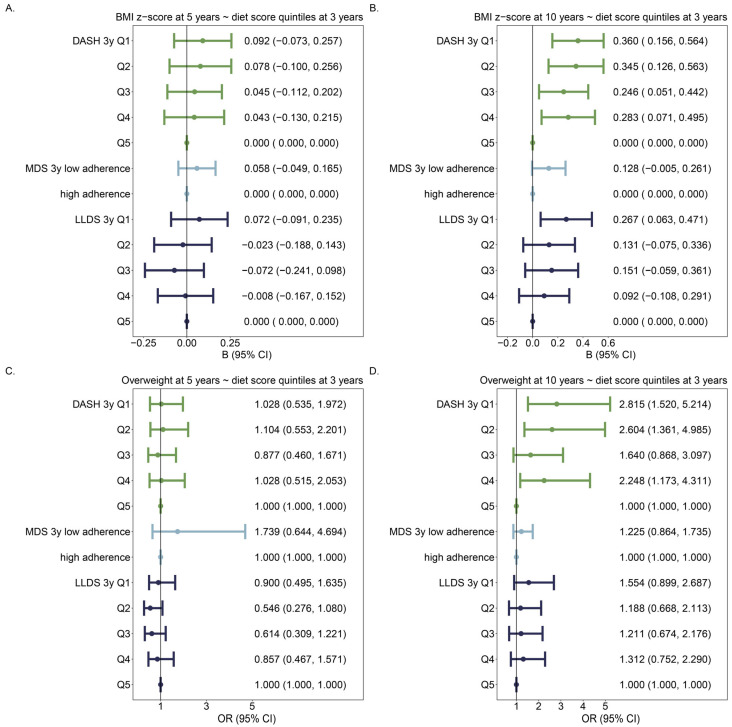
Associations of diet scores in categories with BMI z-score and weight problems including obesity: (**A**–**D**) are the same as Figure 1. Diet scores are expressed as categorical variables in the models, with the best diet quality as the reference category. Associations of each category of diet scores are compared with that of the reference (best) category, adjusting for age, sex, paternal education level, and any smoking during pregnancy. The number of individuals was 971 in 5y models and 959 in 10y models. Different colors identify different diet scores.

**Figure 3 nutrients-16-02634-f003:**
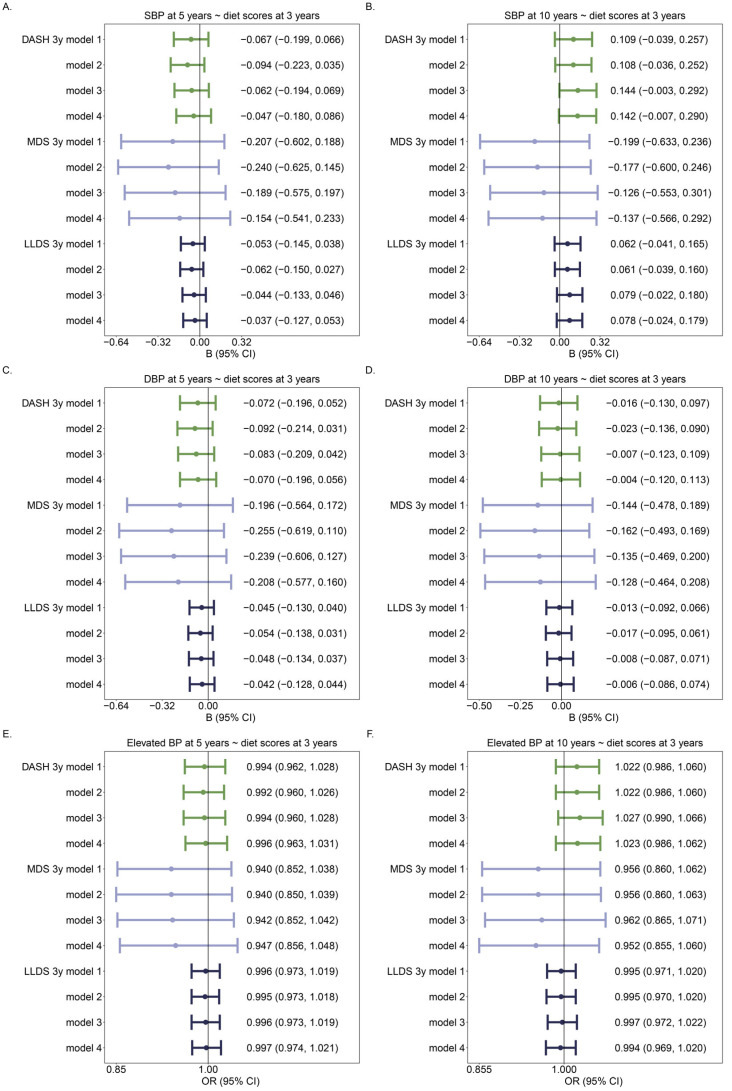
Associations of diet scores (per unit) in continuous with BP and elevated BP: (**A**,**C**) Association between diet scores at 3 years of age and BP at 5 years of age; (**B**,**D**) Association between diet scores at 3 years of age and BP at 10 years of age; (**E**) Association between diet scores at 3 years of age and elevated BP at 5 years of age; (**F**) Association between diet scores at 3 years of age and elevated BP at 10 years of age. For each diet score, there were 4 models: Model 1: Crude model; Model 2: model 1 + age + sex + height; Model 3: model 2 + maternal education level; Model 4: model 3 + any smoking during pregnancy. The number of individuals was 857 in 5y models and 908 in 10y models. Different colors identify different diet scores.

**Figure 4 nutrients-16-02634-f004:**
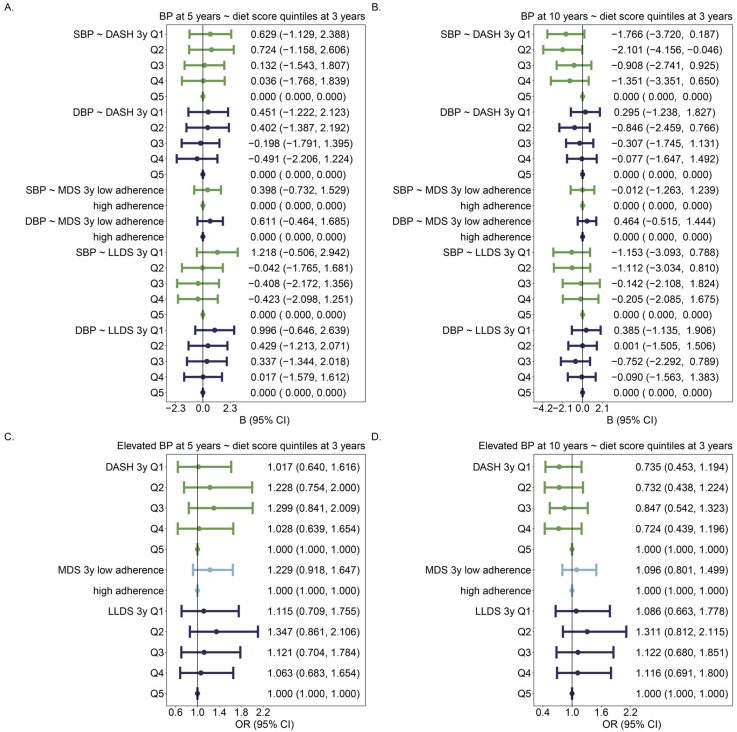
Associations of diet scores in categories with BP and elevated BP: (**A**) Association between diet scores at 3 years of age and BP at 5 years of age; (**B**) Association between diet scores at 3 years of age and BP at 10 years of age; (**C**) Association between diet scores at 3 years of age and elevated BP at 5 years of age; (**D**) Association between diet scores at 3 years of age and elevated BP at 10 years of age. Diet scores are expressed as categorical variables in the models, with the best diet quality as the reference category. Associations of each category of diet scores are compared with that of the reference (best) category, adjusting for age, sex, height, maternal education level, and any smoking during pregnancy. The number of individuals was 857 in 5y models and 908 in 10y models. Different colors identify SBP/DBP (**A**,**B**) or different diet scores (**C**,**D**).

**Figure 5 nutrients-16-02634-f005:**
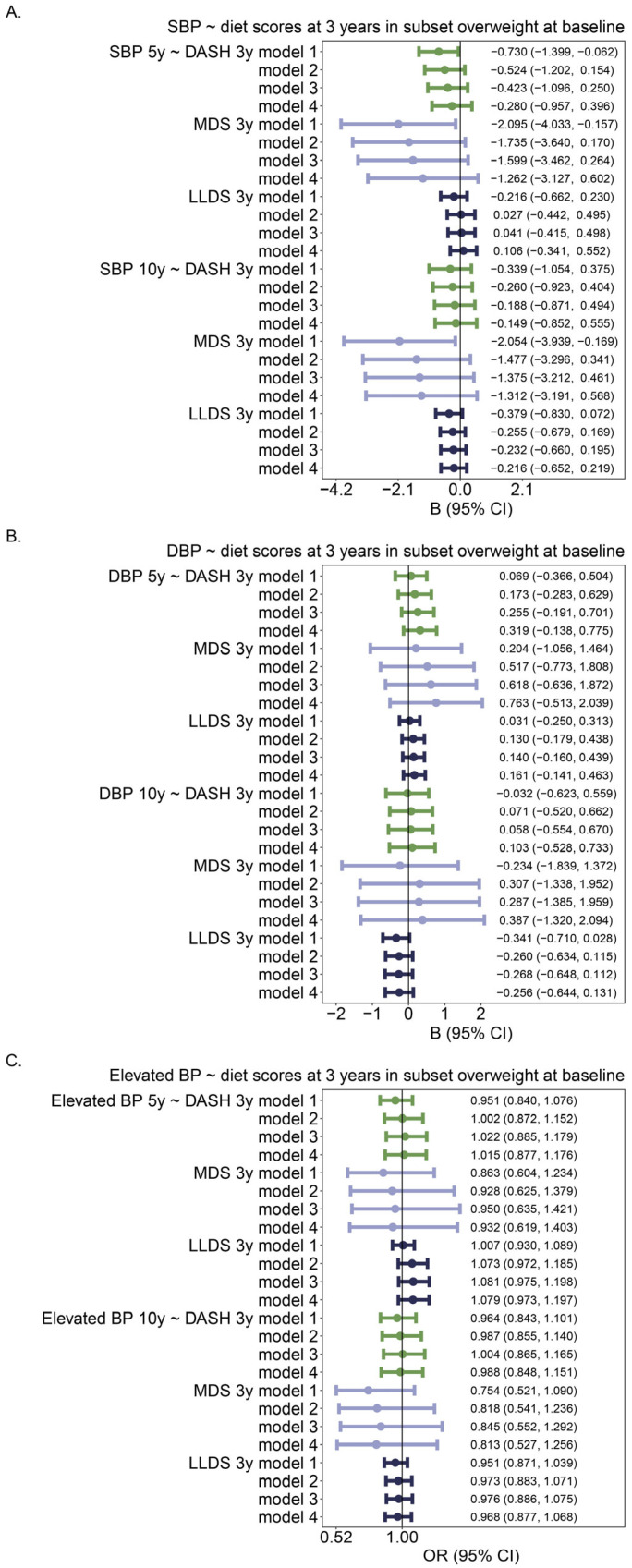
Associations between diet score (per unit) and BP/elevated BP in children who were overweight or obese at baseline (3 years of age). The number of individuals was 62 in the 5y model and 65 in the 10y model. For each diet score, there were 4 models: Model 1: Crude model; Model 2: model 1 + age + sex + height; Model 3: model 2 + maternal education level; Model 4: model 3 + any smoking during pregnancy. (**A**). Association between diet scores at 3 years of age and SBP at 5 and 10 years of age. (**B**). Association between diet scores at 3 years of age and DBP at 5 and 10 years of age; (**C**). Association between diet scores at 3 years of age and elevated BP at 5 and 10 years of age. Different colors identify different diet scores.

**Table 1 nutrients-16-02634-t001:** Participant characteristics.

General Characteristics	N = 1077	
Sex (female), n (%)	524 (48.7%)	
Age at food frequency questionnaire (years), median (IQR)	3.08 (3.02, 3.16)	
Age at outcomes measurement 1 (years), median (IQR)	5.83 (5.58, 6.00)	
Age at outcomes measurement 2 (years), median (IQR)	10.58 (10.25, 10.83)	
Maternal education level within one year of birth, n (%)	High 423 (39.7%)	
	Medium 357 (33.5%)	
	Low 285 (26.8%)	
Paternal education level within one year of birth, n (%)	High 338 (32.4%)	
	Medium 288 (27.6%)	
	Low 416 (39.9%)	
Any smoking during pregnancy, n (%)	118 (11.0%)	
Maternal history of hypertension, n (%)	109 (10.3%)	
Maternal prepregnancy BMI (kg/m^2^), median (IQR)	23.81 (21.61, 26.76)	
Determinants	N = 1077	
DASH score at 3 years of age, median (IQR)	16 (13, 19)	
MDS at 3 years of age, median (IQR)	4 (3, 5)	
LLDS at 3 years of age, median (IQR)	21 (17, 26)	
Outcomes	5 years of age (N = 971)	10 years of age (N = 959)
SBP (mm Hg), median (IQR)	103.33 (98.00, 109.00)	107.19 (101.86, 113.38)
DBP (mm Hg), median (IQR)	61.67 (57.00, 66.33)	63.50 (58.33, 68.00)
Elevated BP (including hypertension), n (%)	293 (34.5%)	217 (23.9%)
Hypertension, n (%)	180 (21.2%)	121 (13.3%)
BMI z-score (SD), median (IQR)	0.21 (−0.31, 0.71)	0.12 (−0.49, 0.91)
Overweight (including obesity), n (%)	105 (10.8%)	158 (16.5%)
Obesity, n (%)	17 (1.8%)	29 (3.0%)

## Data Availability

The data presented in this study are available on request from the corresponding author. The data are not publicly available due to the Confidentiality Statement of Gecko Drenthe Cohort.

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
