# Peer review of "Diet Quality at 3 Years of Age Relates to Lower Body Mass Index but Not Lower Blood Pressure at 10 Years of Age"

_nutrients, 2024, doi:10.3390/nu16162634_

Round 1

Reviewer 1 Report

Comments and Suggestions for Authors

1. Introduction - worthwhile content helps rationalise the focus of the current study throughout.

2. Methods - dietary data has been collected at 3-years of age. While the authors have highlighted the use of a modified scoring system for children of this age range of the Mediterranean Diet Quality Scoring approach, can they either a) add confirmatory detail on the appropriateness of the other scoring systems for 3-year-old children or b) discuss the significant limitations there may be in applying a scoring method to a population group who consume a lower overall amount.

3. Line 169 - the sentence "The association for DASH has repeated in sensitivity analyses..." does not appear to make sense. Please check and revise for clarity.

4. Results - generally, these are presented effectively and annotated appropriately in figures throughout.

5. Line 289 - suggest updating to "The lack of association between..." or similar for clarity.

6. Discussion (BP and other factors) - contextually, it may make sense to consider the range of values in the original dataset when considering presence of absence of associations. A lack of spread/variability in original BP values will mean the chances of seeing an association with dietary habit is low. For BMI-z, based on population estimates in Europe, I would presume a much higher range of possible values, including a higher proportion that were above ideal. Please expand discussion to consider the original dataset, which appears large enough to be nationally representative but has not been designed solely with a view to assess BP and BP changes over time.

7. 

Comments on the Quality of English Language

Some minor issues picked up upon reviewing. Further checking may be required at proofing in the manuscript proceeds

Reviewer 2 Report

Comments and Suggestions for Authors

The manuscript is well-written and addresses an important topic.

I think the biggest problem with the manuscript is that the authors are inconsistent with the use of the terms "overweight" and "overweight or obese."   

Given they analyze for BMI z score and show a similar relationship, I think they are referring to "overweight or obese," but I am not sure.

I think the authors need to clarify their terms.  If they analyzed for just the overweight category, they should analyze the obese category.   I think that should be done regardless.

In the limitations, the authors should point out that food intake was self-reported, and while the food frequency questionnaire is validated, it is still by recall, and the intake, especially for over 1,000 participants, may not be entirely accurate.

Comments on the Quality of English Language

Overall okay.  A few very minor issues.

Round 2

Reviewer 2 Report

Comments and Suggestions for Authors

The authors have successfully addressed all the reviewer concerns.